# Enantioselective total synthesis of (–)-lucidumone enabled by tandem prins cyclization/cycloetherification sequence

Xian-Zhang Liao[1,2,3], Ran Wang[1,2,3], Xin Wang[1,2] & Guang Li [1,2] ✉

The Ganoderma meroterpenoids are a growing class of natural products with architectural complexity, and exhibit a wide range of biological activities. Here, we report an enantioselective total synthesis of the Ganoderma meroterpenoid (–)-lucidumone. The synthetic route features several key transformations, including a) a Cu-catalyzed enantioselective silicon-tethered intramolecular Diels-Alder cycloaddition to construct the highly functionalized bicyclo[2.2.2] octane moiety; b) Brønsted acid promoted tandem O-deprotection/Prins cyclization/Cycloetherification sequence followed by oxidation to install concurrently the tetrahydrofuran and the fused indanone framework; c) Fleming-Tamao oxidation to generate the secondary hydroxyl; d) an iron-catalyzed Wacker-type oxidation of hindered vinyl group to methyl ketone.

Ganoderma fungi (*Ganodermataceae*), a well-known mushroom, has been widely used in traditional Chinese medicine for the prevention and treatment of cancer, hypertension, chronic bronchitis, and asthma for decades[1,2]. The meroterpenoids constituents of *Ganoderma* species have attracted much attention from the pharmacological and synthetic communities due to their diverse and complex structures and potent bioactivities. For example, cochlearol A (**2**)[3–8], cochlearol B (**3**), ganocin A (**4**)[9–14], ganocin B (**5**), and lingzhiol (**6**)[15–24] have been systematically studied by synthetic communities (Fig. 1)[25]. Most recently, an unprecedented cage-like meroterpenoid, namely, lucidumone (**1**), was isolated by Cheng and co-workers from the fruiting bodies of *Ganoderma lucidum* cultivated in Yongsheng County of Yunnan Province in China[26]. Its structure was initially deduced based on extensive 2D-NMR and HRESIMS analysis, featuring a 6/5/6/6/5 polycyclic ring system, one exposed secondary alcohol, a fused indanone motif, and six contiguous stereocenters on the bicyclo[2.2.2]octane subunit. Isolated as a racemate, the two enantiomers were separated by chiral-phase HPLC and the absolute configuration of the (–)-lucidumone was determined by X-ray crystallographic analysis. Preliminary biological evaluation revealed that both enantiomers of **1** exhibit potent inhibitory effects against COX-1 and COX-2. Molecular docking study revealed that (–)-lucidumone (–)-(**1**) selectively inhibits COX-2 via bonding to Tyr385

and Ser530 residues, making it a potential lead in searching for anti-inflammatory drugs. Undoubtedly, the structural complexity and the potential biological activity render it an attractive synthetic target. In 2022, She and co-workers reported the construction of a pentacyclic skeleton of lucidumone (**1**) in racemic form featuring an elegant oxidative dearomatization/intramolecular Diels−Alder cycloaddition and acid-promoted dynamic kinetic resolution cyclization[27]. Torre's group accomplished the first total synthesis of (+)-lucidumone involving an enantioselective inverse-electron-demand Diels−Alder reaction (IEDDA) and a one pot retro-[4 + 2]/[4 + 2] cycloaddition sequence[28]. Most recently, Kawamoto and Ito's group disclosed a fascinating total synthesis of racemic lucidumone (**1**) by means of one-pot Claisen rearrangement/intramolecular aldolization for the construction of the tetracyclic framework. The enantioselective total synthesis of (–)-lucidumone (–)-(**1**) was also accomplished through a chiral transfer strategy in the Claisen rearrangement step[29]. In this report, we disclose an enantioselective total synthesis of (–)-lucidumone (–)-(**1**). The salient features of our strategy include a Cu-catalyzed enantioselective silicon-tethered intramolecular Diels-Alder cycloaddition to assemble the bicyclo[2.2.2]octane framework and a domino deprotection/Prins reaction/Cycloetherification/oxidation sequence to generate concurrently the tetrahydrofuran and the fused indanone skeleton.

[1]State Key Laboratory of Bioactive Substance and Function of Natural Medicines, Institute of Materia Medica, Chinese Academy of Medical Sciences & Peking Union Medical College, 100050 Beijing, P. R. China. [2]Beijing Key Laboratory of Active Substance Discovery and Druggability Evaluation, Institute of Materia Medica, Chinese Academy of Medical Sciences & Peking Union Medical College, 100050 Beijing, P. R. China. [3]These authors contributed equally: Xian-Zhang Liao, Ran Wang. ✉e-mail: guang.li@imm.ac.cn

## Results

With an in-depth analysis of the chemical structure of lucidumone, we envisioned that the critical tetrahydrofuran-spiro-indanone framework (E-B-A rings of the natural product), a challenging task for synthesis, could be constructed via a pivotal cascade Prins cyclization/Cycloetherification followed by oxidation[30,31]. The precursor for this key transformation could be divided into two parts, the left bicyclo[2.2.2]octane moiety (highlighted in red) and right benzene (highlighted in blue), which would be connected through the transition metal-catalyzed C7′–C3 bond formation. Owing to the high regio- and stereoselectivity of intramolecular Diels–Alder cycloaddition than the intermolecular one, the silicon-tethered intramolecular Diels–Alder

reaction gives us an excellent choice for the assembly of bicyclo[2.2.2]octane moiety[32-35]. In addition to the expected high degrees of regio- and stereoselectivity, the silicon-tethered intramolecular Diels−Alder reaction could also provide suitable synthetic handles for the rapid generation of the requisite secondary alcohol and methyl ketone.

### Retrosynthetic analysis

With the hypothesis in mind, we proposed our retrosynthetic analysis of (−)-lucidumone (−)-**1** in Fig. 2. We envisioned to access (−)-**1** from hexacyclic intermediate **7** via a sequence of Fleming−Tamao oxidation, Wacker oxidation and *O*-demethylation. The fused indanone framework could be constructed by a key *O*-deprotection/Prins reaction/Cycloetherification followed by oxidation sequence from **8**, which in turn could be assembled via a Suzuki coupling between boronic acid **9** and vinyl triflate **10**. The latter could be traced back to **11**, the product of asymmetric intramolecular Diels–Alder reaction of **12**, which could be accessed from the known diene **14**[36] and the easily accessible silyl acrylate imide **13**.

### Synthesis of chiral bicyclo[2.2.2]octane

The synthesis of **1** commenced with the preparation of **12** (Fig. 3). Following the protocol developed by Sieburth[37], exposure of the readily available silyl acrylate imide **13** to triflic acid, followed by the addition of the known primary alcohol **14** afforded silyl ether **12** in 91% yield. The Diels–Alder cycloaddition of **12** in the presence of Et₂AlCl (0.1 equiv) at 0 °C proceeded smoothly to give the desired cycloadduct in 90% yield as a single *endo* diastereoisomer. Encouraged by this observation, we turned our attention to the catalytic asymmetric silicon-tethered intramolecular Diels–Alder reaction of **12**[38,39]. Following Evans's pioneering work, diverse chiral $C_2$-symmetric Box ligands were used in combination with different Cu(II) salt[40-42]. Screening the reaction

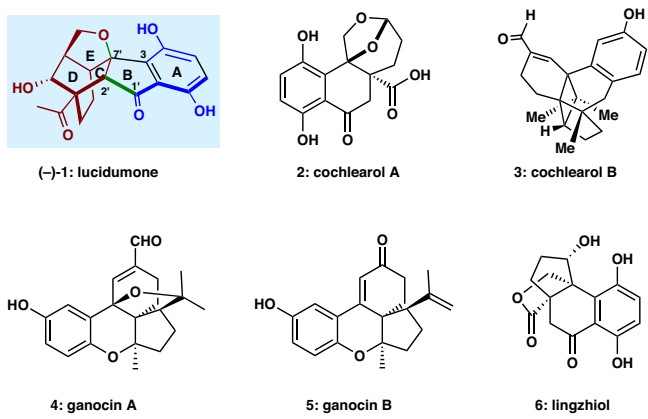

**Fig. 1 | Representative *Ganoderma* meroterpenoids.** (−)-lucidumone [(−)-(**1**)], cochlearol A (**2**), cochlearol B (**3**), ganocin A (**4**), ganocin B (**5**), and lingzhiol (**6**).

**Fig. 2 | Retrosynthetic analysis of (−)-lucidumone [(−)-1].** The key disconnections involve Cu-catalyzed enantioselective silicon-tethered intramolecular Diels−Alder cycloaddition to construct C/D rings and a tandem Prins cyclization/Cycloetherification sequence to construct rings E/B/A, respectively.

conditions by varying the reaction temperature, the ligands, and the copper sources led to the identification of the optimized reaction condition using Cu(OTf)$_2$ (10 mol%) and ligand (S,S)-**L1** (11 mol%) at 50 °C (see Supplementary Table 1). Under these conditions, cycloadduct **11** was isolated in 96% yield with 92% *ee*. Performing the reaction at 3.0 mmol in the presence of Cu(OTf)$_2$ (0.3 mmol), (S,S)-**L1** (0.33 mmol) afforded **11** without obvious erosion of yield and enantioselectivity (95% yield, 92% *ee*). The *ee* value of **11** could be further improved to 99% after

recrystallization. The absolute configuration of **11** was determined by the X-ray diffraction analysis of its derivative (vide infra, compound **17**).

### Synthesis of hexacyclic intermediate

With the gram-scale synthesis of chiral bicyclo[2.2.2]octane intermediate **11** in hand, the synthesis of highly functionalized hexacyclic intermediate **22** was undertaken (Fig. 4). Hydrolysis of **11** (LiOH, H$_2$O$_2$, THF/H$_2$O v/v = 2:1, 0 °C to RT) afforded the carboxylic acid **15** in 94%

| [Cu] | Ligand | Yield | ee |
|---|---|---|---|
| Cu(SbF$_6$)$_2$ | L1 | 88 | 92 |
| Cu(OTf)$_2$ | L1 | 96 | 92 |
| Cu(OTf)$_2$ | L2 | 90 | 85 |
| Cu(OTf)$_2$ | L3 | 86 | 63 |

**Fig. 3 | Synthesis of chiral bicyclo[2.2.2]octane 11.** Reagents and conditions: **a** TfOH (2.2 equiv), 2,6-lutidine (2.4 equiv), DCM, –78 to 0 °C, then **14**, 91%; **b** Cu(OTf)$_2$ (0.1 equiv), **L1** (0.11 equiv), DCM, 50 °C, 96% yield, 92% *ee*. TfOH triflic acid, 2,6-lutidine 2,6-dimethylpyridine, DCM dichloromethane.

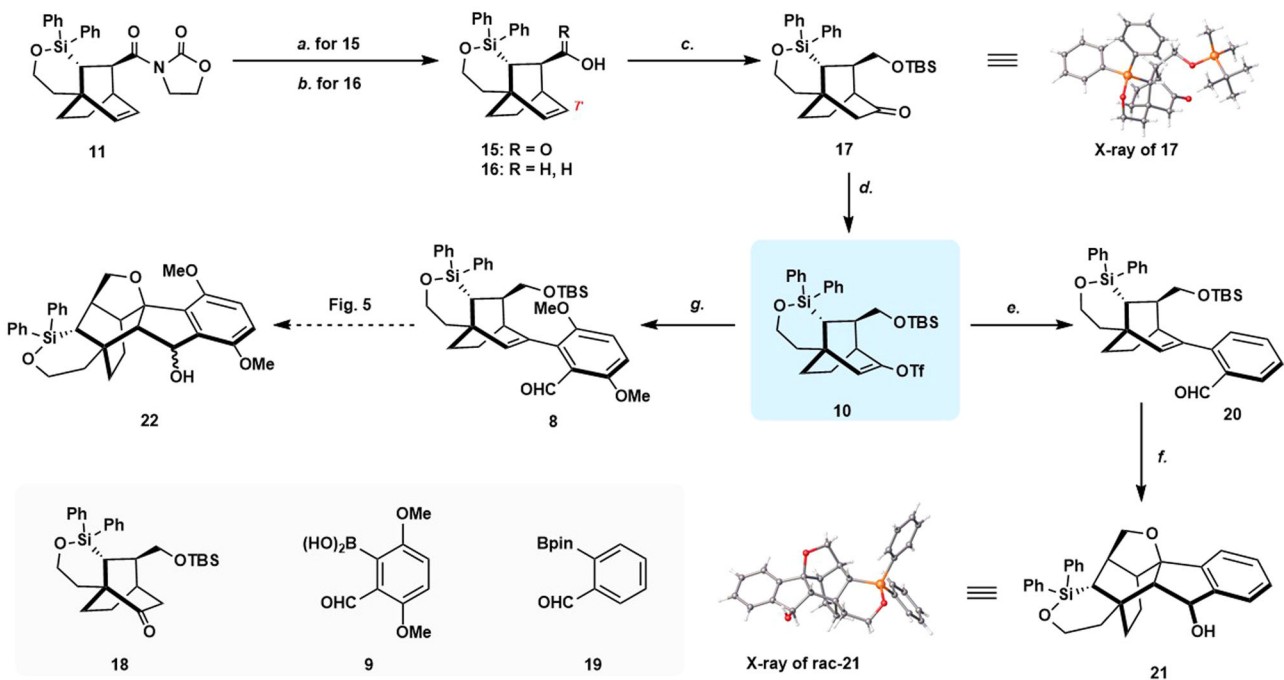

**Fig. 4 | Synthesis of hexacyclic intermediate 22.** Reagents and conditions: **a** LiOH (3.0 equiv), H$_2$O$_2$ (8.8 equiv), THF-H$_2$O (v:v = 2:1), 0 °C to RT, 94%; **b** LiBH$_4$ (4.0 equiv), THF, 0 °C, 85%; **c** i. BH$_3$·THF (5.0 equiv), THF, –30 °C to RT, NaBO$_3$·4H$_2$O (10.0 equiv). ii. TBSCl (3.0 equiv), imid (5.0 equiv), DCM, RT. iii. DMP (2.0 equiv), NaHCO$_3$ (10 equiv), RT, 69% for **17**, 20% for **18**; **d** KHMDS (1.4 equiv), PhNTf$_2$ (1.4 equiv), THF, –78 °C, 94%; **e 19** (2.0 equiv), Pd(dppf)Cl$_2$ (0.1 equiv), K$_3$CO$_3$ (3.0 equiv), DMSO, 80 °C, 96%; **f** HCl (2 M in ethyl acetate) (10.0 equiv), DCM, –10 °C, 89%; **g 9** (2.0 equiv), Pd(dppf)Cl$_2$ (0.1 equiv), S-Phos (0.2 equiv), K$_3$PO$_4$ (3.0 equiv), DMF, 80 °C, 69%. THF tetrahydrofuran, TBSCl *tert*-Butyldimethylsilyl chloride, imid Imidazole, DMP Dess–Martin periodinane, KHMDS potassium bis(trimethylsilyl) amide, PhNTf$_2$ N-phenyl-bis(trifluoromethanesulfonimide), S-Phos 2-dicyclohexylphosphino-2′,6′-dimethoxybiphenyl, DMF N, N-dimethylformamide.

yield. On the other hand, reduction of **11** with lithium borohydride (LiBH$_4$) provided the alcohol **16** in 85% yield. We surmised that the neighboring carboxylic acid group of **15** or the hydroxy group of **16** could be used to direct the hydroboration at C7[43,44]. Treatment of **15** with BH$_3$·THF (–30 °C to RT.) followed by the addition of NaBO$_3$·4H$_2$O furnished the mixed diol product resulting from the concomitant reduction of the carboxylic acid. Without purification, this diol was converted to ketone **17** and its regioisomer **18** in 89% overall yield (rr = 3.4:1) via a sequence of selective *O*-protection and oxidation. The absolute configuration of the tricycle ketone **17** was determined by X-ray crystallographic analysis. Therefore, the successful elaboration of **17** according to our synthetic plan would afford the levorotatory target molecular (–)-lucidumone (–)-**1**. Notably, applying the same conditions to alcohol **16**, a mixture of two regioisomeric ketones was isolated in 1:1 ratio. These results indicated that carboxylic acid is a better-directing group in the hydroboration step.

We then turned our attention to build A-B-E rings of the natural product. Deprotonation of ketone **17** with potassium bis(trimethylsilyl) amide (KHMDS) at –78 °C followed by the addition of PhNTf$_2$ afforded the desired vinyl triflate **10** in 94% isolated yield. The Pd-catalyzed Suzuki–Miyaura cross-coupling between **10** and the known boronic acid

**9**[45] proceeded smoothly to afford the functionalized tetracycle **8** in 69% yield. The one-step conversion of **8** to **22**, a strategic step in our synthesis, was sought next. This efficient transformation involved a hypothetical acid-catalyzed *O*-deprotection/Prins cyclization/Cycloetherification sequence[30,31,46–53]. Model reaction was attempted first before the real substrate. The precursor **20**, generated through Suzuki–Miyaura cross-coupling between **10** and commercially available 2-formylphenylboronic acid pinacol ester **19**, was screened in various acidic conditions for the key transformation. Gratifyingly, the desirable **21** was obtained in the presence of HCl (10 equiv) in DCM at –10 °C in 89% yield with 20:1 diastereoselectivity (X-ray of major rac-**21**). However, tetracycle **8** is unstable and was decomposed when treated with the aforementioned identical condition. The realization of the transformation turned out to be a challenging task. The desired product as a pair of diastereoisomers **22** could be isolated under a series of acidic conditions with the concomitant formation of Prins cyclization product **23**. To minimize the generation of by-products, different Brønsted acids [TFA, HCl, *p*-TsOH·H$_2$O, (–)-CSA] and Lewis acids [BF$_3$·Et$_2$O, AlCl$_3$, Sc(OTf)$_3$, TBSOTf, TMSOTf, etc.] in different solvents at various temperatures were screened (Fig. 5). HCl was still identified as the most suitable promotor. The optimum reaction conditions consisted of

| Entry[a] | Acid (equiv) | Temp. (°C) | Yield of 22 (%) | Yield of 23 (%) | Yield of 24 (%) | Yield of 25 (%) |
|---|---|---|---|---|---|---|
| 1 | HCl (10 eq) | –10 °C | - | - | - | - |
| 2 | BF$_3$·Et$_2$O(1.0 eq) | –78 °C | 19 | 42 | - | - |
| 3 | TBSOTf (1.0 eq) | –78 °C | - | 18 | trace | - |
| 4 | TMSOTf (1.0 eq) | –78 °C | trace | - | - | 53 |
| 5 | AlCl$_3$ (1.0 eq) | –78 °C | 23 | 35 | - | - |
| 6 | Sc(OTf)$_3$ (1.0 eq) | –78 °C | 15 | 54 | - | - |
| 7 | (-)-CSA (1.0 eq) | –78 °C | - | 85 | - | - |
| 8 | *p*-TSA·H$_2$O (1.0 eq) | –78 °C | trace | 65 | - | - |
| 9 | TFA (1.0 eq) | –78 °C | trace | 53 | - | - |
| 10 | HCl (1.0 eq) | –78 °C | 12 | 75 | - | - |
| 11 | HCl (10.0 eq) | –78 °C | 86 | - | - | - |
| 12 | Silica gel | rt | - | 47 | - | - |

**Fig. 5 | Optimization of tandem O-deprotection/Prins cyclization/Cycloetherification conditions.** [a]Conditions: **8** (0.03 mmol), DCM (1.5 mL. *c* 0.02 M), N$_2$ atmosphere.

stirring a DCM solution of **8** in the presence of an excess of HCl (2 M in ethyl acetate, 10 equiv) at –78 °C for 24 h (Fig. 5, Entry 11). Under these conditions, **22** was generated as a pair of diastereoisomers in 86% yield, which could also be directly oxidized by Dess−Martin periodinane to afford the desired product **7** in a one-pot sequence in 80% overall yield (Fig. 6). The structure and absolute configuration of **7** was confirmed by X-ray crystallographic analysis. We note that the reaction temperature and time have to be carefully controlled in order to avoid the formation of by-products.

### Total synthesis of (–)-lucidumone

Elaboration of hexacycle **7** to (–)-lucidumone (–)-**1** was accomplished as shown in Fig. 6. The Fleming−Tamao oxidation of **7** (KF, H₂O₂, methanol/THF, v/v = 1:1) afforded diol **26** in 91% yield[54]. It was noted that by-product **23** could also be converted to diol **26** under the sequential Dess−Martin oxidation and the Fleming−Tamao oxidation via a spontaneous deprotection/oxa-Michael addition. Dehydration of **26** underwent smoothly to give the desired product **27** (X-ray) in 50% isolated yield according to Grieco's procedure[55]. As Kawamoto and Ito mentioned[29], the late-stage Wacker oxidation of the vinyl group **27** to methyl ketone group **28** proved to be challenging. Under classic conditions, aldehyde resulting from the *anti*-Markovnikov addition was formed as a major product due presumably to the steric effect[56]. We then turned our attention to the iron-catalyzed Wacker-type oxidation of olefins to ketones developed by Han's[57] and Knölker's[58] groups, respectively. Gratefully, applying their standard conditions to **27** [Fe(dbm)₃, PhSiH₃, EtOH, RT], methylketone **28** was obtained in 65%

yield. Finally, Lewis acid catalyzed double O-demethylation of **28** afforded (–)-lucidumone (–)-**1** in 70% isolated yield. The physical and spectroscopic data of the synthetic (–)-lucidumone (–)-**1** are identical to those reported for the natural product.

In this work, an enantioselective total synthesis of (–)-lucidumone (–)-**1** is accomplished. Our synthetic route to this natural product features several key transformations, including bis(oxazoline)copper(II) complex catalyzed enantioselective silicon-tethered intramolecular Diels−Alder reaction to construct the multi-functionalized bicyclo[2.2.2]octane moiety, a Brønsted acid promoted tandem O-deprotection/Prins reaction/Cycloetherification followed by oxidation to install concurrently the tetrahydrofurane and the fused indanone moieties, Fleming−Tamao oxidation to afford the secondary hydroxyl, and an iron-catalyzed late-stage Wacker-type oxidation of hindered vinyl group to methyl ketone.

## Methods

Unless otherwise stated, reagents were purchased at the highest commercial quality and used without further purification. Solvents were purchased in HPLC quality, degassed by purging thoroughly with nitrogen, and dried over activated molecular sieves of appropriate size. Alternatively, they were purged with argon and passed through alumina columns in a solvent purification system (Innovative Technology). The conversion was monitored by thin layer chromatography (TLC) using Merck TLC silica gel 60 F254. Compounds were visualized by UV light at 254 nm and by dipping the plates in an ethanolic vanillin/sulfuric acid solution or a 72 aqueous potassium permanganate solution followed by

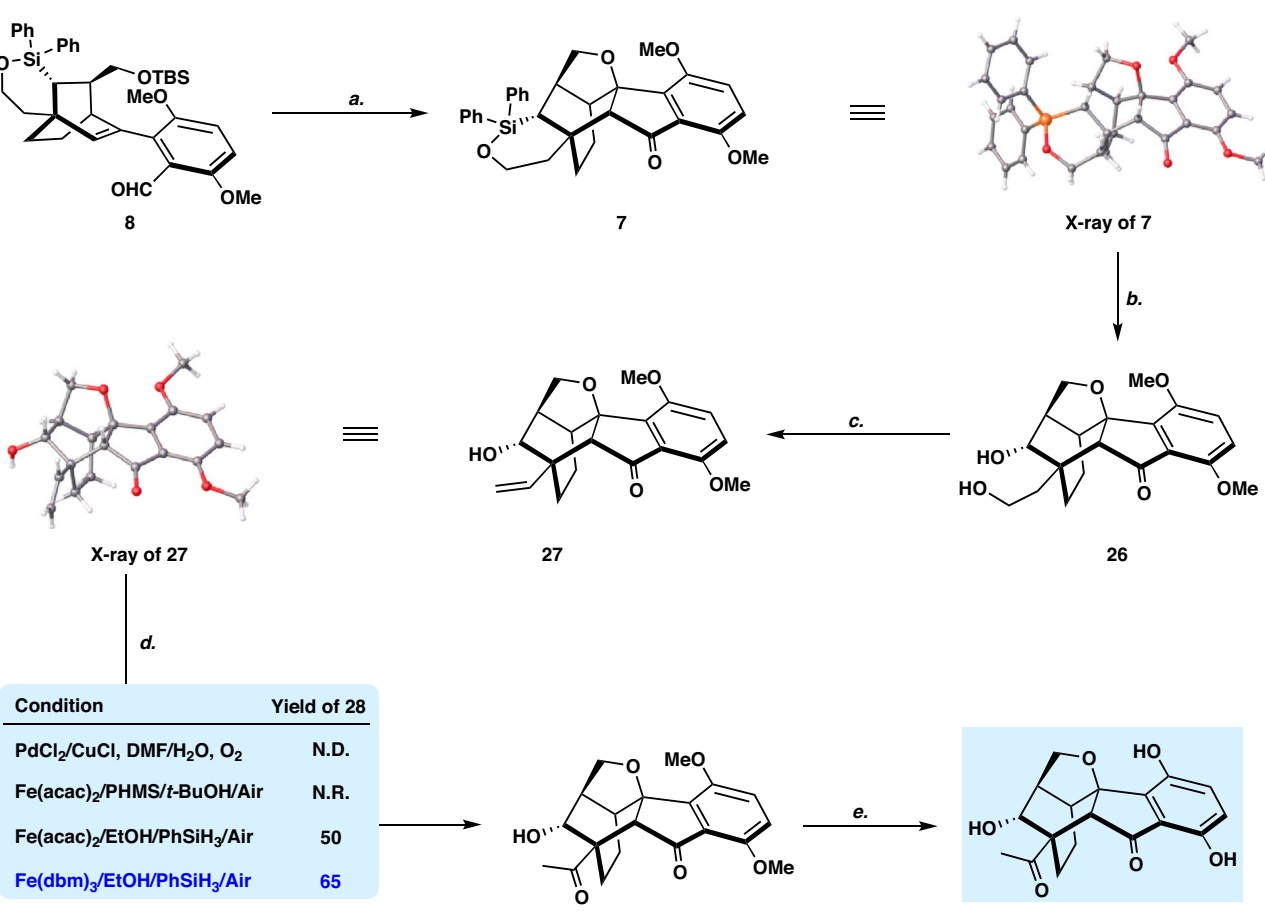

| Condition | Yield of 28 |
|---|---|
| PdCl₂/CuCl, DMF/H₂O, O₂ | N.D. |
| Fe(acac)₂/PHMS/*t*-BuOH/Air | N.R. |
| Fe(acac)₂/EtOH/PhSiH₃/Air | 50 |
| **Fe(dbm)₃/EtOH/PhSiH₃/Air** | **65** |

**Fig. 6 | Total synthesis of (–)-lucidumone [(–)−1].** Reagents and conditions: **a** HCl (2 M in ethyl acetate) (10.0 equiv), DCM, –78 °C, then DMP (2.0 equiv), NaHCO₃ (10.0 equiv), RT, 80%; **b** KF (5.0 equiv), H₂O₂ (20 equiv), KHCO₃ (1.6 equiv), MeOH- THF (v:v = 1:1), 50 °C, 91%; **c** 2-nitrophenylselenocyanate (1.2 equiv), Py (1.2 equiv), PBu₃ (1.2 equiv), H₂O₂ (24 equiv), THF, 50%; **d** Fe(dbm)₃ (0.1 equiv), PhSiH₃ (10 equiv), EtOH, RT, 65%; **e** AlCl₃ (10 equiv), 1-dodecanethiol (20 equiv), DCM, RT, 70%.

heating. Flash column chromatography was performed over silica gel (230–400 mesh).

NMR spectra were recorded on a Bruker 500 and 400 MHz at room temperature, Chemical shifts ($\delta$) were reported in parts per million (ppm) relative to residual solvent peaks rounded to the nearest 0.01 for proton and 0.1 for carbon. Coupling constants ($J$) were reported in Hz to the nearest 0.1 Hz. Peak multiplicity was indicated as follows s (singlet), d (doublet), t (triplet), q (quartet), m (multiple), and br (broad). The attribution of peaks was done using the multiplicities and integrals of the peaks.

IR spectra were recorded in a Perkin-Elmer 1000 series FT-IR spectrometer. The spectra were reported in $cm^{-1}$.

The accurate masses were measured by the mass spectrometry service of IMM, PUMC&CAMS on an Agilent 6244 Tof-MS using electrospray ionization (ESI).

## Data availability

The X-ray crystallographic coordinates for structures reported in this study have been deposited at the Cambridge Crystallographic Data Centre (CCDC) under deposition numbers 2271397 (**17**), 2277937 (**7**), 2312931 (**21**), and 2284054 (**27**). Copies of the data can be obtained free of charge via https://www.ccdc.cam.ac.uk/structures/. All other data supporting the findings of this study, including experimental procedures and compound characterization, NMR, and HPLC, are available within the Article and its Supplementary Information or from the corresponding author upon request.

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

## Acknowledgements

Financial support for this work was provided by the National Natural Science Foundation of China (Grant No. 22371301 to G.L.), Non-profit Central Research Institute Fund of Chinese Academy of Medical Sciences (2021-RC350-005 to G.L.), and startup funding (to G.L.) from State Key Laboratory of Bioactive Substance and Function of Natural Medicines, Institute of Materia Medica, Chinese Academy of Medical Sciences & Peking Union Medical College. We also thank Prof. Jieping Zhu (Ecole Polytechnique Fédérale de Lausanne) and Prof. Liansuo Zu (Tsinghua University) for helpful discussions and advice during the preparation of this manuscript.

## Author contributions

X.-Z.L., R.W., X.W., and G.L. conceived and designed the experiments. G.L. directed the project. X.-Z.L. and R.W. carried out the experiments. X.-Z.L., R.W., X.W., and G.L. interpreted the results. G.L. wrote the manuscript.

## Competing interests

The authors declare no competing interests.
