## [Peer Review File · Nature Communications]

Enantioselective Total Synthesis of (–)-Lucidumone Enabled by Tandem Prins Cyclization/Cycloetherification SequenceREVIEWERS' COMMENTS

Reviewer #1 (Remarks to the Author):

Li and coworkers reported an enantioselective total synthesis of the (–)-lucidumone. The synthetic route features an intramolecular Diels-Alder cycloaddition, a tandem O-deprotection/Prins cyclization/cycloetherification sequence, a Fleming-Tamao oxidation and an iron-catalyzed Wacker-type oxidation. The writing is clear and the Supporting Information is of high quality. This work could be published in *Nat. Commun.* after address the following issues.

(1) Some spelling errors should be corrected:

a) Page 3 in Figure 2, – I think the word “sequencean” in the statement should be changed to “sequence”.

b) In SI, the word “rection” should be changed to “reaction” in the synthetic procedures of compounds 13, 17, 20, 8, 26, 27.

(2) A new review about total synthesis of Ganoderma Meroterpenoids by Kawamoto and Ito has just been accepted (*Asian J. Org. Chem.* 2024, e202300633), which should be included.

(3) Page 4 in Figure 3, – In the table, when using Cu(OTf)₂ and L1, compound 11 can be obtained in 96% yield. However, in the statement, the yield is 95%. The data should be consistent.

(4) Page 6 in Table 1, – I would like to know whether the author has attempted to increase the dosage of other acids in the tandem O-deprotection/Prins cyclization/cycloetherification.

(5) Page 7 in Figure 5 and Page 18 in SI, – The abbreviation of dibenzoylmethane is “dbm” not “dam”.

(6) Page 9 in SI, – In the data of compound 16, “¹³C NMR (125 MHz, CDCl₃)” should be changed to “¹³C NMR (100 MHz, CDCl₃)”

Reviewer #2 (Remarks to the Author):

This manuscript describes an enantioselective total synthesis of the meroterpenoid lucidumone. The approach proposed by the authors involves a silicon-tethered enantioselective intramolecular Diels-Alder cycloaddition as the key enantiodetermining step, as well as Prins cyclization / cycloetherification cascade to assemble the pentacyclic skeleton. Overall, this elegant approach brings some interesting lessons regarding the cascade Prins/cycloetherification, the late Wacker oxidation (including unexpected regioselectivity issues) and the silicon-tethered intramolecular Diels-Alder strategy. Two enantioselective syntheses of the same natural product have been published previously (*JACS* 2022 and *Angewandte* 2023). In comparison to these previous approaches, the synthesis reported here has a similar step count (13 steps LLS in the *JACS* 2022, 14 steps in the *Angewandte* 2023, 14 steps LLS from S1 here) and similar overall yield. Despite the elegant approach and scholarly nature of the work, I am not convinced, in light of the literature precedents, that the synthesis detailed herein warrants publication in a journal of the calibre of *Nat. Commun.*

Reviewer #3 (Remarks to the Author):

In this manuscript, the authors described total synthesis of (–)-lucidumone. Key reactions

include copper (II) complex catalyzed enantioselective silicon-tethered intramolecular Diels-Alder reaction, Suzuki-Miyaura coupling, Prince reaction/cycloetherification sequence, Fleming-Tamao oxidation, and Wacker-type oxidation. Two total syntheses and one synthesis of the pentacyclic structure have already been reported by another groups. However, the authors employed different synthetic strategy, presenting some new chemistry that would attract the synthetic community. Therefore, this manuscript would deserve publication in Nature Communication after minor revision of some grammatical and lexical mistakes.

Ms Title: Enantioselective Total Synthesis of (–)-Lucidumone Enabled by Tandem Prins Cyclization/Cycloetherification Sequence” (MS N^o: NCOOMS-24-04908).

MS N^o: NCOOMS-24-04908

Point-by-point responses to referees' comments

Reviewer #1 (Remarks to the Author):

Li and coworkers reported an enantioselective total synthesis of the (–)-lucidumone. The synthetic route features an intramolecular Diels-Alder cycloaddition, a tandem O-deprotection/Prins cyclization/cycloetherification sequence, a Fleming-Tamao oxidation and an iron-catalyzed Wacker-type oxidation. The writing is clear and the Supporting Information is of high quality. This work could be published in Nat. Commun. after address the following issues.

We would like to thank you for the positive and insightful comments on the manuscript. We are very grateful to you for reviewing the paper so carefully.

(1) Some spelling errors should be corrected:

a) Page 3 in Figure 2, – I think the word “sequencean” in the statement should be changed to “sequence”.

It has been corrected in the revised version.

b) In SI, the word “rection” should be changed to “reaction” in the synthetic procedures of compounds 13, 17, 20, 8, 26, 27.

It has been corrected in the revised SI.

(2) A new review about total synthesis of Ganoderma Meroterpenoids by Kawamoto and Ito has just been accepted (*Asian J. Org. Chem.* 2024, e202300633), which should be included.

This reference was added in the revised version.

Ref 25. Kawamoto, Y., Ito, H. Total Synthesis of Ganoderma Meroterpenoids – Progresses since 2014. *Asian J. Org. Chem.* e202300633 (2024).

(3) Page 4 in Figure 3, – In the table, when using Cu(OTf)₂ and L1, compound 11 can be obtained in 96% yield. However, in the statement, the yield is 95%. The data should be consistent.

It has been corrected in the revised version.

(4) Page 6 in Table 1, – I would like to know whether the author has attempted to increase the dosage of other acids in the tandem O-deprotection/Prins cyclization/cycloetherification.

Thanks very much for your question. Multiple equivalents of other acids were attempted for this transformation, but no improvement for positive results, and some acids even led to the decomposition of the substrate.

(5) Page 7 in Figure 5 and Page 18 in SI, – The abbreviation of dibenzoylmethane is “dbm” not “dam”.

It has been corrected in the revised version and SI

(6) Page 9 in SI, – In the data of compound 16, “¹³C NMR (125 MHz, CDCl₃)” should be changed to “¹³C NMR (100 MHz, CDCl₃)”

The ¹H NMR and ¹³C NMR spectra of compound **16** were tested in 400 MHz and 500MHz NMR machine, respectively. So the data of ¹³C NMR spectra should be (100 MHz, CDCl₃)”.

Reviewer #2 (Remarks to the Author):

This manuscript describes an enantioselective total synthesis of the meroterpenoid lucidumone. The approach proposed by the authors involves a silicon-tethered enantioselective intramolecular Diels-Alder cycloaddition as the key enantiodetermining step, as well as Prins cyclization / cycloetherification cascade to assemble the pentacyclic skeleton. Overall, this elegant approach brings some interesting lessons regarding the cascade Prins/cycloetherification, the late Wacker oxidation (including unexpected regioselectivity issues) and the silicon-tethered intramolecular Diels-Alder strategy.

Two enantioselective syntheses of the same natural product have been published previously (JACS 2022 and Angewandte 2023). In comparison to these previous approaches, the synthesis reported here has a similar step count (13 steps LLS in the JACS 2022, 14 steps in the Angewandte 2023, 14 steps LLS from S1 here) and similar overall yield. Despite the elegant approach and scholarly nature of the work, I am not convinced, in light of the literature precedents, that the synthesis detailed herein warrants publication in a journal of the calibre of Nat. Commun.

We would like to thank you for the positive and insightful comments on the manuscript. We are very grateful to you for the time and effort in reviewing this manuscript.

Reviewer #3 (Remarks to the Author):

In this manuscript, the authors described total synthesis of (-)-lucidumone. Key reactions include copper (II) complex catalyzed enantioselective silicon-tethered intramolecular Diels-Alder reaction, Suzuki-Miyaura coupling, Prince reaction/cycloetherification sequence, Fleming-Tamao oxidation, and Wacker-type oxidation. Two total syntheses and one synthesis of the pentacyclic structure have already been reported by another groups. However, the authors employed different synthetic strategy, presenting some new chemistry that would attract the synthetic community. Therefore, this manuscript would deserve publication in Nature Communication after minor revision of some grammatical and lexical mistakes.

We would like to thank you for the valuable and insightful comments on the manuscript. We are very grateful to you for reviewing the paper so carefully.

We have re-checked the whole content and corrected all potential grammatical errors and misspellings in this revised manuscript.